# Scalable Diffusion for Materials Generation

**Mengjiao Yang**[†,◇]**, KwangHwan Cho**[†]**, Amil Merchant**[◇]**, Pieter Abbeel**[†]**,**
**Dale Schuurmans**[◇,‡]**, Igor Mordatch**[◇]**, Ekin Dogus Cubuk**[◇]
[†]UC Berkeley, [◇]Google DeepMind, [‡]University of Alberta
sherryy@{berkeley.edu, google.com}
unified-materials.github.io

## Abstract

Generative models trained on internet-scale data are capable of generating novel and realistic texts, images, and videos. A natural next question is whether these models can advance science, for example by generating novel stable materials. Traditionally, models with explicit structures (e.g., graphs) have been used in modeling structural relationships in scientific data (e.g., atoms and bonds in crystals), but generating structures can be difficult to scale to large and complex systems. Another challenge in generating materials is the mismatch between standard generative modeling metrics and downstream applications. For instance, common metrics such as the reconstruction error do not correlate well with the downstream goal of discovering *novel* stable materials. In this work, we tackle the scalability challenge by developing a unified crystal representation that can represent *any* crystal structure (UniMat), followed by training a diffusion probabilistic model on these UniMat representations. Our empirical results suggest that despite the lack of explicit structure modeling, UniMat can generate high fidelity crystal structures from larger and more complex chemical systems, outperforming previous graph-based approaches under various generative modeling metrics. To better connect the generation quality of materials to downstream applications, such as discovering novel stable materials, we propose additional metrics for evaluating generative models of materials, including per-composition formation energy and stability with respect to convex hulls through decomposition energy from Density Function Theory (DFT). Lastly, we show that conditional generation with UniMat can scale to previously established crystal datasets with up to millions of crystals structures, outperforming random structure search (the current leading method for structure discovery) in discovering new stable materials.

## 1 Introduction

Large generative models trained on internet-scale vision and language data have demonstrated exceptional abilities in synthesizing highly realistic texts [1, 2], images [3, 4], and videos [5, 6]. The need for novel synthesis, however, goes far beyond conversational agents or generative media, which mostly impact the digital world. In the physical world, technological applications such as catalysis [7], solar cells [8], and lithium batteries [9] are enabled by the discovery of novel materials. The traditional trial-and-error approach that discovered these materials can be highly inefficient and take decades (e.g., blue LEDs [10] and high-Tc superconductors [11]). Generative models have the potential to dramatically accelerate materials discovery by generating and evaluating material candidates with desirable properties more efficiently in silico.

One of the difficulties in materials generation lies in characterizing the structural relationships between atoms, which scales quadratically with the number of atoms. While representations with explicit structures such as graphs have been extensively studied [12, 13, 14, 15], explicit characterization of inter-atomic relationships becomes increasingly challenging as the number of atoms increases, which can prevent these methods from scaling to large materials datasets with complex chemical

AI for Science Workshop at NeurIPS 2023.

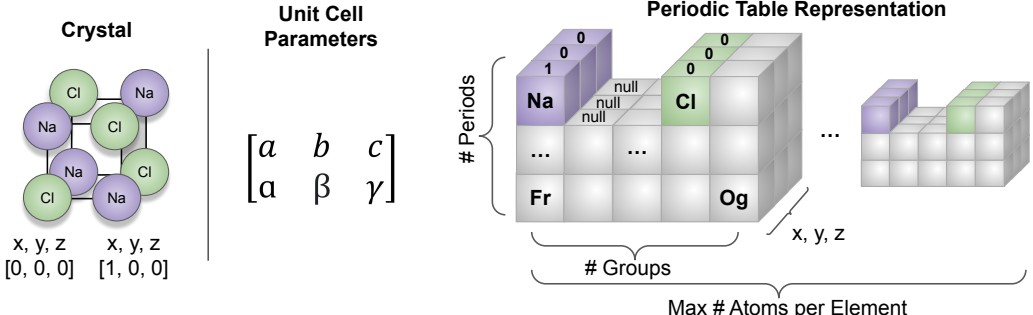

Figure 1: UniMat representation of crystal structures. Crystals are represented by the atom locations stored at the corresponding elements in the periodic table (and additional unit cell parameters if coordinates are fractional). For instance, the bottom right atom Na in the crystal is located at $[1, 0, 0]$, hence the periodic table has value $[1, 0, 0]$ at the Na entry.

systems. On the other hand, given that generative models are designed to discover patterns from data, it is natural to wonder if material structures can automatically arise from data through generative modeling, similar to how natural language structures arise from language modeling, so that large system sizes becomes more of a benefit than a roadblock.

Existing generative models that directly model atoms without explicit structures are largely inspired by generative models for computer vision, such as learning VAEs or GANs on voxel images [16, 17] or point cloud representations of materials [18]. VAEs and GANs have known drawbacks such as posterior collapse [19] and mode collapse [20], potentially making scaling difficult [21]. More recently, diffusion models [22, 23] have been found particularly effective in generating diverse yet high fidelity image and videos, and have been applied to data at internet scale [24, 5]. However, it is unclear whether diffusion models are also effective in modeling structural relationships between atoms in crystals that are neither images nor videos.

In this work, we investigate whether diffusion models can capture inter-atomic relationships effectively by directly modeling atom locations, and whether such an approach can be scaled to complex chemical systems with a larger number of atoms. Specifically, we propose a unified representation of materials (UniMat) that can capture *any* crystal structure. As shown in Figure 1, UniMat represents atoms in a material's unit cell (the smallest repeating unit) by storing the continuous value $x, y, z$ atom locations at the corresponding element entry in the periodic table. This representation overcomes the difficulty around joint modeling of discrete atom types and continuous atom locations. With such a unified representation of materials, we train diffusion probabilistic models by treating the UniMat representation as a 4-dimensional tensor and applying interleaved attention and convolution layers, similar to [24], across periods and groups of the periodic table. This allows UniMat to capture inter-atom relationships while preserving any inductive bias from the periodic table, such as elements in the same group having similar chemical properties.

We first evaluate UniMat on a set of proxy metrics proposed by [15], and show that UniMat generally works better than the previous state-of-the-art graph based approach and a recent language model baseline [25]. However, we are ultimately interested in whether the generated materials are physically valid and can be synthesized in a laboratory. In answering this question, we run DFT relaxations [26] to compute the formation energy of the generated materials, which is more widely accepted in material science than learned proxy metrics in [27]. We then use per-composition formation energy and stability with respect to convex hull through decomposition energy as more reliable metrics for evaluating generative models for materials. UniMat drastically outperforms previous state-of-the-art according to these DFT based metrics.

Lastly, we scale UniMat to train on all experimentally verified stable materials as well as additional stable / semi-stable materials found through search and substitution (over 2 million structures in total). We show that predicting material structures conditioned on element type can generalize (in a zero-shot manner) to predicting more difficult structures that are not a neighboring structure to the training set, achieving better efficiency than the predominant random structure search. This allows for the possibility of discovering new materials with desired properties effectively. In summary, our work contributes the following:

- We develop a novel representation of materials that enables diffusion models to scale to large and complex materials datasets, outperforming previous methods on previous proxy metrics.
- We conduct DFT calculations to rigorously verify the stability of generated materials, and propose to use per-composition formation energy and stability with respect to convex hull for evaluating generative models for materials.
- We scale conditional generation to all known stable materials and additional materials found by search and substitution, and observe zero-shot generalization to generating harder structures, achieving better efficiency than random structure search in discovering new materials.

## 2 Scalable Diffusion for Materials Generation

We start by proposing a novel crystal representation that can represent any material with a finite number of atoms in a unit cell (the smallest repeating unit of a material). We then illustrate how to learn both unconditional and conditional denoising diffusion models on the proposed crystal representations. Lastly, we explain how we can verify generated materials rigorously using quantum mechanical methods.

### 2.1 Scalable Representation of Crystal Structures

An ideal representation for crystal structures should not introduce any intrinsic errors (unlike voxel images), and should be able to support both up scaling to large sets of materials on the internet and down scaling to a single compound system that a particular group of scientists care about (e.g., silicon carbide). We develop such a scalable and flexible representation below.

**Periodic Table Based Material Representation.** We first observe that periodic table captures rich knowledge of chemical properties. To introduce such prior knowledge to a generative model as an inductive bias, we define a 4-dimensional material space, $\mathcal{M} := \mathbb{R}^{L \times H \times W \times C}$, where $H = 9$ and $W = 18$ correspond to the number of periods and groups in the periodic table, $L$ corresponds to the maximum number of atoms per element in the periodic table, and $C = 3$ corresponds to the x,y,z locations of each atoms in a unit cell. We define a *null* location using special values such as $x = y = z = -1$ to represent the absence of this atom. A visualization of this representation is shown in Figure 1. To account for invariances in order, rotation, translation, and periodicity, we incorporate data augmentation through random shuffling and rotations similar to [28, 18, 29]. Note that when crystals are represented using Cartesian coordinates, this representation is already sufficient for expressing any crystal structure $x \in \mathcal{M}$ with less than $L$ atoms per chemical element. When crystals are represented using fractional coordinates, we need additional unit cell parameters $(a, b, c) \in \mathbb{R}^3$ and $(\alpha, \beta, \gamma) \in \mathbb{R}^3$ to specify the lengths and angles between edges of the unit cell as shown in Figure 1. We denote this representation UniMat, as it is a unified representation of crystals, and has the potential to represent broader chemical structures (e.g., drugs, molecules, and proteins).

**Flexibility for Smaller Systems.** While UniMat can represent any crystal structure, sometimes one might only be interested in generating structures with one specific element (e.g., carbon in graphene) or two-chemical compounds (e.g., silicon carbide). Instead of setting $H$ and $W$ to the full periods and groups of the periodic table, one can set $H = 1, W = 1$ (for one specific element) or $H = 9, W = 2$ (for elements from two groups) to model specific chemical systems of interest. $L$ can also be adjusted according to the number of elements expected to exist in the system.

### 2.2 Learning Diffusion Models with UniMat Representation

With the UniMat representation above, we now illustrate how effective training of diffusion models [30, 23] on crystal structures can be enabled, followed by how to generate crystal structures conditioned on compositions or other types of material properties. Details of the model architecture and training procedure can be found in Appendix 6.

**Diffusion Model Background.** Denoising diffusion probablistic models are a class of probabilistic generative models initially designed for images where the generation of an image $x \in \mathbb{R}^d$ is formed by iterative denoising. That is, given an image $x$ sampled from a distribution of images $p(x)$, a randomly sampled Gaussian noise variable $\epsilon \sim \mathcal{N}(0, I_d)$, and a set of $T$ different noise levels $\beta_t \in \mathbb{R}$, a denoising model $\epsilon_\theta$ is trained to denoise the noise corrupted image $x$ at each specified noise level $t \in [1, T]$ by minimizing:

$$\mathcal{L}_{\text{MSE}} = \|\epsilon - \epsilon_\theta(\sqrt{1 - \beta_t}x + \sqrt{\beta_t}\epsilon, t))\|^2.$$

Given this learned denoising function, new images may be generated from the diffusion model by initializing an image sample $x_T$ at noise level $T$ from a Gaussian $\mathcal{N}(0, I_d)$. This sample $x_T$ is then

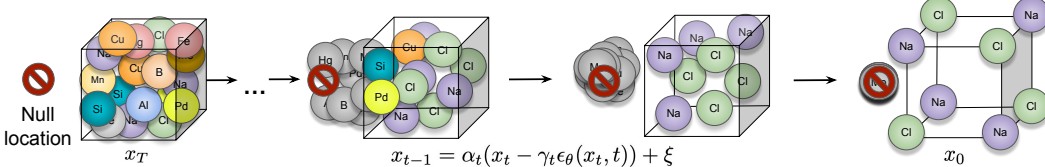

Figure 2: Illustration of the denoising process for unconditional generation with UniMat. The denoising model learns to move atoms from random locations back to their original locations. Atoms not present in the crystal are moved to the null location during the denoising process, allowing crystals with an arbitrary number of atoms to be generated.

iteratively denoised by following the expression:

$$x_{t-1} = \alpha_t(x_t - \gamma_t \epsilon_\theta(x_t, t)) + \xi, \quad \xi \sim \mathcal{N}(0, \sigma_t^2 I_d), \tag{1}$$

where $\gamma_t$ is the step size of denoising, $\alpha_t$ is a linear decay on the currently denoised sample, and $\sigma_t$ is some time varying noise level that depends on $\alpha_t$ and $\beta_t$. The final sample $x_0$ after $T$ rounds of denoising corresponds to the final generated image.

**Unconditional Diffusion with UniMat.** Now instead of an image $x \in \mathbb{R}^d$, we have a material $x \in \mathbb{R}^d$ with $d = L \times H \times W \times 3$ tensor as described in Section 2.1, where the inner-most dimension of $x$ represents the atom locations (x,y,z). The denoising process in Equation 1 now corresponds to the process of moving atoms from random locations back to their original locations in a unit cell as shown in Figure 2. Note that the set of null atoms (i.e., atoms that do not exist in a crystal) will have random locations initially (left-most structure in Figure 2), and are gradually moved to the special null location during the denoising process. The null atoms are then filtered when the final crystals are extracted. The inclusion of null atoms in the representation enables UniMat to generate crystals with an arbitrary number of atoms (up to a maximum size). We parametrize $\epsilon_\theta(x_t, t)$ using interleaved convolution and attention operations across the $L, H, W$ dimensions of $x_t$ similar to [24], which can capture inter-atom relationships in a crystal structure. When atom locations are represented using fractional coordinates, we treat unit cell parameters as additional inputs to the diffusion process by concatenating the unit cell parameters with the crystal locations.

**Conditioned Diffusion with UniMat.** While the unconditional generation procedure described above allows generation of materials from random noise, the learned materials distribution $p(x)$ would largely overlap with the training distribution. This is undesirable in the context of materials discovery, where the goal is to discover *novel* materials that do not exist in the training set. Futhermore, practical applications such as material synthesis often focus on specific types of materials, but one do not have much control over what compound gets generated during an unconditional denoising process. This suggests that conditional generation may be more relevant for materials discovery.

We consider conditioning generation on compositions (types and ratios of chemical elements) $c \in \mathbb{R}^{H \times W}$ when only the composition types are specified (e.g., carbon and silicon), or on $c \in \mathbb{R}^{L \times H \times W}$ when the exact composition (number of atoms per element) is given (e.g., Si4C4). We denote the conditional denoising model as $\epsilon_\theta(x_t, t | c)$. Since the input to the unconditional denoising model $\epsilon_\theta(x_t, t)$ is a noisy material of dimensions $(L, H, W, 3)$, we concatenate the conditioning variable $c$ with the noisy material along the last dimension before inputting the noisy material into the denoising model, so that the denoising model can easily condition on compositions as desired.

In addition to conditioning on compositions, one may also want to incorporate material properties or information such as formation energy, bandgap, or even textual descriptions into the generation process. Since conditioning on this auxiliary information does not have to be enforced strictly, similar to composition conditioning, we can leverage classifier-free guidance [31] and use

$$\hat{\epsilon}_\theta(x_t, t | c, \texttt{aux}) = (1 + \omega)\epsilon_\theta(x_t, t | c, \texttt{aux}) - \omega\epsilon_\theta(x_t, t | c) \tag{2}$$

as the denoising model in the reverse process for sampling materials conditioned on auxiliary information aux, where $\omega$ controls the strength of auxiliary information conditioning.

## 2.3 Evaluating Generated Materials

Different from generative models for vision and language where the quality of generation can be easily assessed by humans, evaluating generated crystals rigorously requires calculations from Density Functional Theory (DFT) [32], which we elaborate in detail below.

**Drawbacks of Learning Based Evaluations.** One way to evaluate generative models for materials is to compare the distributions of formation energy $E_f$ between a generated and reference

set, $D(p(E_f^{\text{gen}}), p(E_f^{\text{ref}}))$, where $D$ is a distance measure over distributions, such as earth mover's distance [15]. Since using DFT to compute $E_f$ is computationally demanding, previous work has relied on a learned network to predict $E_f$ from generated materials [15]. However, predicting $E_f$ can have intrinsic errors, particularly in the context of materials discovery where the goal is to generate *novel* materials beyond the training manifold of the energy prediction network.

Even when $E_f$ can be predicted with reasonable accuracy, a low $E_f$ does not necessarily reflect ground-truth (DFT) stability. For example, [27] reported that a model that can predict $E_f$ with an error of 60 meV/atom (a 16-fold reduction from random-guessing) does not provide any predictive improvement over random guessing for stable material discovery. This is because most variations in $E_f$ are between different chemical systems, whereas for stability assessment, the important comparison is between compounds in a single chemical system. When materials generated by two different models contain different compounds, the model that generated materials with a lower $E_f$ could have simply generated compounds from a lower $E_f$ system without enabling efficient discovery [33].

The property that captures *relative* stabilities between different compositions is known as decomposition energy ($E_d$). Since $E_d$ depends on the formation energy of other compounds from the same system, predicting $E_d$ directly using machine learning models has been found difficult [27].

**Evaluating via Per-Composition Formation Energy.** Different from learned energy predictors, DFT calculations provide more accurate and reliable $E_f$ values. When two models each generate a structure of the same composition, we can directly compare which structure has a lower DFT computed $E_f$ (and is hence more stable). We call this the *per-composition* formation energy comparison. We define average difference in per-composition formation energy between two sets of materials $A$ and $B$ as

$$\Delta E_f(A, B) = \frac{1}{|C|} \sum_{(x,x') \in C} \left( E_{f,x}^A - E_{f,x'}^B \right), \tag{3}$$

where $C = \{(x, x') \mid x \in A, x' \in B, \text{comp}(x) = \text{comp}(x')\}$ denotes the set of structures from $A$ and $B$ that have the same composition. We also define the $E_f$ Reduction Rate between set A and B as the rate where structures in A have a lower $E_f$ than the structures in B of the corresponding compositions, i.e.,

$$E_f \text{ Reduction Rate}(A, B) = \frac{1}{|C|} |\{(x, x') \mid (x, x') \in C \wedge E_{f,x}^A < E_{f,x'}^B\}|, \tag{4}$$

where $C$ is the same as in Equation 3. We can then use $\Delta E_f$ and the $E_f$ Reduction Rate to compare a generated set of structures to some reference set, or to compare two generated sets. $\Delta E_f(A, B)$ measures how much lower in $E_f$ (on average) the structures in a set $A$ are compared to the structures of correponding compositions in a set $B$, while $E_f$ Reduction Rate$(A, B)$ reflects how many structures in $A$ have lower $E_f$ than the corresponding structures in $B$. We use these metrics to evaluate generated materials in Section 3.2.1.

**Evaluating Stability via Decomposition Energy** We also want to compare generated materials that differ in composition. To do so, we can use DFT to compute decomposition energy $E_d$. $E_d$ measures a compound's thermodynamic decomposition enthalpy into its most stable compositions on a convex hull phase diagram, where the convex hull is formed by linear combinations of the most stable (lowest energy) phases for each known composition [34]. As a result, decomposition energy allows us to compare compounds from two generative models that differ in composition by separately computing their decomposition energy with respect to the convex hull formed by a larger materials database. The distribution of decomposition energies will reflect a generative model's ability to generate relatively stable materials. We can further compute the number of *novel* stable ($E_d < 0$) materials from set $A$ with respect to convex hull as

$$\# \text{Stable}(A) = |\{x \in A \mid E_{d,x}^A < 0\}|, \tag{5}$$

and compare this quantity to some other set $B$. We apply this metric to evaluate generative models for materials in Section 3.2.

**Evaluating against Random Search Baseline.** For structure prediction given compositions, one popular non-learning based approach is Ab initio random structure search (AIRSS) [35]. AIRSS works by initializing a set of sensible structures given the composition and a target volume, relaxing randomly initialized structures via soft-sphere potentials, followed by DFT relaxations to minimize the total energy of the system. However, discovering structures (especially if done in a high-

| Method | Dataset | Validity % ↑ | | COV % ↑ | | Property Statistics ↓ | | |
| --- | --- | --- | --- | --- | --- | --- | --- | --- |
| | | Structure | Composition | Recall | Precision | Density | Energy | # Elements |
| CDVAE | Perov5 | 100 | 98.5 | 99.4 | 98.4 | 0.125 | 0.026 | 0.062 |
| | Carbon24 | 100 | – | 99.8 | 83.0 | 0.140 | 0.285 | – |
| | MP20 | **100** | 86.7 | 99.1 | 99.4 | 0.687 | 0.277 | 1.432 |
| LM | Perov5 | 100 | 98.7 | **99.6** | **99.4** | **0.071** | – | 0.036 |
| | MP20 | 95.8 | 88.8 | 99.6 | 98.5 | 0.696 | – | 0.092 |
| UniMat | Perov5 | **100** | 98.8 | 99.2 | 98.2 | 0.076 | **0.022** | **0.025** |
| | Carbon24 | **100** | – | **100** | 96.5 | 0.013 | 0.207 | – |
| | MP20 | 97.2 | **89.4** | **99.8** | **99.7** | 0.088 | 0.034 | 0.056 |

Table 1: Proxy evaluation of unconditional generation using CDVAE [15], language model [25], and UniMat. UniMat generally performs better in terms of property statistics, and achieves the best coverage on more difficult dataset (MP-20). We note the limitation of these proxy metrics, and defer more rigorous evaluation to DFT calculations.

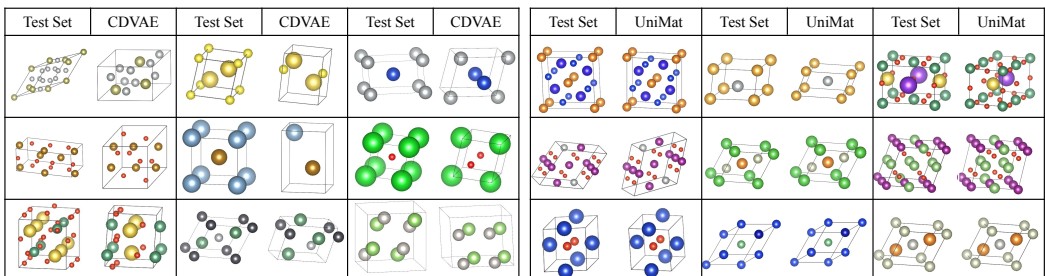

Figure 3: Qualitative evaluation of materials generated by CDVAE [15] (left) and UniMat (right) trained on MP-20 in comparison to the test set materials of the same composition. Materials generated by UniMat generally align better with the test set.

throughput framework) requires a large number of initializations and relaxations which can often fail to converge [36, 33].

One practical use of conditional UniMat is to propose initial structures given compositions, with the hope that the generated structures will result in a higher convergence rate for DFT calculations compared to structures proposed by AIRSS, which are based on manual heuristics and random guessing of initial volumes. We can further conduct formation and decomposition energy analysis similar to evaluating unconditional generations on structures proposed by AIRSS and generative models.

## 3 Experimental Evaluation

We now evaluate UniMat using both the previous proxy metrics from [15] as well as metrics derived from DFT calculations, as discussed in Section 2.3. UniMat is able to generate orders of magnitude more stable materials verified by DFT calculations compared to the previous state-of-the-art generative model. We further demonstrate UniMat's ability in accelerating random structure search through conditional generation.

### 3.1 Evaluating Unconditional Generation Using Proxy Metrics

**Datasets, Metrics, and Baselines.** We begin the evaluation following the same setup as CD-VAE [15], and train three generative models on Perov-5, Carbon-24, and MP-20 materials datasets. We report metrics on structural and composition validity determined by atom distances and SMACT, coverage metrics based on CrystalNN fingerprint distances, and property distributions in density, learned formation energy, and number of atoms following CDVAE. In addition to CDVAE, we include a recent language model baseline that learns to directly generate crystal files [25].

**Results.** Evaluation results on UniMat and baselines are shown in Table 5. All three models perform similarly in terms of structure and composition validity on the Perov-5 dataset due to its simplicity. UniMat performs slightly worse on the coverage based metrics on Perov-5, but achieves better distributions in energy and number of unique elements. On Carbon-24, UniMat outperforms CDVAE in all metrics. On the more realistic MP-20 dataset, UniMat achieves the best property

statistics, coverage, and composition validity, but worse structure validity than CDVAE. Results on full coverage metrics from CDVAE are in Appendix 9.

In addition, we qualitatively evaluate the generated materials from training on MP-20 in Figure 3. We select generated materials that have the same composition as the test set from MP-20, and use the VESTA crystal visualization tool [37] to plot both the test set materials and the generated materials. The range of fractional coordinates in the VESTA settings were set from -0.1 to 1.1 for all coordinates to represent all fractional atoms adjacent to the unit cell. In general, we found that UniMat generates materials that are visually more aligned with the test set materials than CDVAE.

**Ablation on Model Size.** In training on larger datasets with more diverse materials such as MP-20, we found benefits in scaling up the model as shown in Table 4, which suggests that the UniMat representation and the UniMat training objective can be further scaled to systems larger than MP-20, which we elaborate more in Section 3.3.

|  | Validity % ↑ |  | COV % ↑ |  |
|---|---|---|---|---|
| Model size | Struct. | Comp. | Recall | Precision |
| Small (64) | 95.7 | 86.0 | 99.8 | 99.3 |
| Medium (128) | 96.8 | 86.7 | 99.8 | 99.5 |
| Large (256) | **97.2** | **89.4** | **99.8** | **99.7** |

Figure 4: UniMat trained with a larger feature dimension results in better validity and coverage.

## 3.2 Evaluating Unconditional Generation Using DFT Calculations

As discussed in Section 2.3, proxy-based evaluation in Section 3.1 should be backed by DFT verifications similar to [16]. In this section, we evaluate stability of generated materials using metrics derived from DFT calculations in Section 2.3.

### 3.2.1 Per-Composition Formation Energy

**Setup.** We start by running DFT relaxations using the VASP software [26] to relax both atomic positions and unit cell parameters on generated materials from models trained on MP-20 to compute their formation energy $E_f$ (see details of DFT in Appendix 7). We then compare average difference in per-composition formation energy ($\Delta E_f$ in Equation 3) and the formation energy reduction rate ($E_f$ Reduction Rate in Equation 4) between materials generated by CDVAE and the MP-20 test set, between UniMat and the test set, and between UniMat and CDVAE.

**Results.** We plot the difference in formation energy for each pair of generated structures from UniMat and CDVAE with the same composition in Figure 5. We see the majority of the generated compositions from UniMat have a lower formation energy. We further report $\Delta E_f$ and the $E_f$ Reduction Rate in Table 2. We see that among the set of materials generated by UniMat and CDVAE with overlapping compositions, 86% of them have a lower energy when generated by UniMat. Furthermore, materials generated by UniMat have an average of -0.21 eV/atom lower $E_f$ than CDVAE. Comparing the generated set against the MP-20 test set also favors UniMat.

### 3.2.2 Stability Analysis through Decomposition Energy

As discussed in Section 2.3, generated structures relaxed by DFT can be compared against the convex hull of a larger materials database in order to analyze their stability through decomposition energy. Specifically, we downloaded the full Materials Project database [34] from July 2021, and used this to form the convex hull. We then compute the decomposition energy for materials generated by UniMat and CDVAE individually against the convex hull.

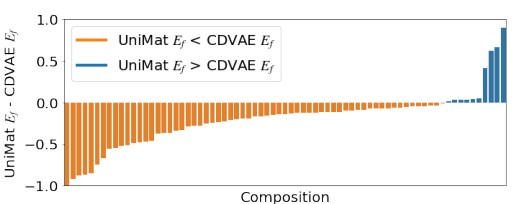

Figure 5: Difference in $E_f$ for each composition generated by UniMat and CDVAE, i.e., $E_{f,x}^A - E_{f,x'}^B$, where $A$ and $B$ are sets of structures generated by UniMat and CDVAE, respectively. UniMat generates more structures with lower $E_f$.

| A, B | $\Delta E_f$ (eV/atom) | $E_f$ Reduc. Rate |
|---|---|---|
| CDVAE, MP-20 test | 0.279 | 0.083 |
| UniMat, MP-20 test | **0.061** | **0.254** |
| UniMat, CDVAE | **-0.216** | **0.863** |

Table 2: $\Delta E_f$ (Equation 3) and $E_f$ Reduction Rate (Equation 4) between CDVAE and MP-20 test, between UniMat and MP-20 test, and between UniMat and CDVAE. UniMat generates structures with an average of -0.216 eV/atom lower $E_f$ than CDVAE. 86.3% of the overlapping (in composition) structures generated by UniMat and CDVAE has a lower energy in UniMat.

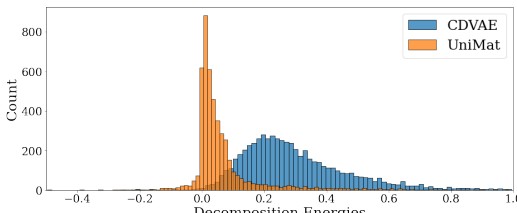

Figure 6: Histogram of decomposition energy $E_d$ of structures generated by CDVAE and UniMat after DFT relaxation. UniMat generates structures with lower decomposition energies.

| | # Stable MP 2021 | # Metastable MP 2021 | # Stable GNoME |
|---|---|---|---|
| CDVAE | 56 | 90 | 1 |
| UniMat | **414** | **2157** | **32** |

Table 3: Number of stable ($E_d < 0$) and metastable ($E_d < 25$meV/atom) materials generated compared against the convex hull of MP 2021, and stability against GNoME with 2 million structures. UniMat generates an order of magnitude more stable / metastable materials than CDVAE.

| Ba2TbIr1O6 | CsCeSe2 | ErBi2ClO4 | KI10 | KTmTe2 | KGdSe2 | MgBr10 | Rb2TcF6 | Sm2Cl2O2 | Sr2BrN |
|---|---|---|---|---|---|---|---|---|---|

Figure 7: Visualizations of materials generated by UniMat trained on MP-20 before DFT relaxation that have $E_d < 0$ after relaxation compared against the convex hull of MP 2021. We note that these materials require further analysis and verification before they can be claimed to be realistic or stable.

**Results.** We plot the distributions of the decomposition energies after DFT relaxation for the generated materials from both models in Figure 6. Note that only the set of generated materials that converged after DFT calculations are plotted. We see that UniMat generates materials that are lower in decomposition energy after DFT relaxation compared to CDVAE. We further report the number of newly discovered stable / metastable materials (with $E_d < 25$meV/atom) from both UniMat and CDVAE in Table 3. In addition to using the convex hull from Materials Project 2021, we also use another dataset (GNoME) with 2.2 million materials constructed via structure search to construct a more challenging convex hull [33]. We see that UniMat is able to discover an order of magnitude more stable materials than CDVAE with respect to convex hulls constructed from both datasets. We visualize examples of newly discovered stable materials by UniMat in Figure 7.

### 3.3 Evaluating Composition Conditioned Generation

We have verified that some of the unconditionally generated materials from UniMat are indeed novel and stable through DFT calculations. We now assess composition conditioned generation which is often more practical for downstream synthesis applications.

**Setup.** For the structure search baseline, we use AIRSS to randomly initialize 100 structures per composition for a fixed set of compositions followed by relaxation via soft-sphere potentials. We then run DFT relaxations on these AIRSS structures. For conditional generation using UniMat, we train composition conditioned UniMat (as described in Section 2.2) on the GNoME dataset consisting of 2.2 million stable materials. We then sample 100 structures per composition for the same set of compositions used by AIRSS. We then evaluate the rate of compositions for which at least 1 out of 100 structures converged during DFT calculations for both structures initialized by AIRSS and by UniMat. In addition to convergence rate, we also evaluate the $\Delta E_f$(UniMat, AIRSS) and the $E_f$ Reduction Rate (UniMat, AIRSS) on the DFT relaxed structures. Since none of the test compositions exist in the training set of GNoME, we are essentially evaluating the ability of UniMat to generalize to more difficult structures in a zero-shot manner. See the detailed setup of AIRSS in Appendix 8.

**Results.** We first observe that AIRSS has an overall convergence rate of 0.55, whereas UniMat has an overall convergence rate of 0.81. We note that both AIRSS and UniMat can be further optimized for convergence rate, so these results are only initial signals on how conditional generative models compare to structure search. Next, we take the relaxed structure with the lowest $E_f$ from both UniMat and AIRSS for each composition, and plot the per-composition $E_f$ difference in Figure 8, and $\Delta E_f$(UniMat, AIRSS) $= -0.68$eV/atom, and

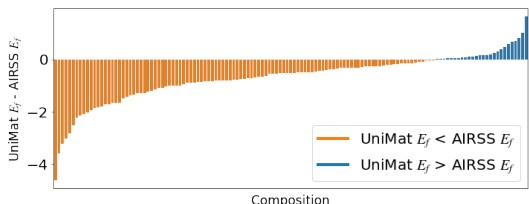

Figure 8: Difference in per-composition formation energy between structures produced by UniMat and AIRSS. More compounds generated by UniMat lead to lower formation energy than AIRSS.

$E_f$ Reduction Rate(UniMat, AIRSS) $= 0.8$, which suggests that UniMat is indeed effective in initializing structures that lead to lower $E_f$ than AIRSS.

## 4   Related Work

**Diffusion Models for Structured Data**   Diffusion models [22, 23, 38] were initially proposed for generating images from noise of the same dimension through a Markov chain of Gaussian transitions, and have been adopted to structured data such as graphs [39, 40, 41, 42], sets [43] and point clouds [44, 45, 46]. Diffusion modeling for materials requires joint modeling of continuous atom locations and discrete atom types. Previous approaches either embed discrete quantities into a continuous latent space, risking information loss [15], or directly learn discrete-space transformations [40, 47] on graphs represented by adjacency matrices that scale quadratically in the number of atoms.

**Generative Models for Materials Discovery.**   Generative models originally designed for images have been applied to generating material structures, such as GANs [48, 18, 49], VAEs [28, 16, 50, 29], and diffusion models [15]. These methods were developed to work with different materials representations as voxel images [28, 16, 29], graphs [15], point clouds [18], and phase fields or electron density maps [51, 29]. However, existing work has mostly focused on simpler materials in binry compounds [16, 49], ternary compounds [48, 18], or cubic systems [28]. [15] show that graph neural networks with latent space diffusion guided by gradient of formation energy can scale to larger materials datasets such as the Materials Project [34]. However, the quality of generated materials seems to decrease drastically when scaled to larger systems. Recently, large language models have been applied to directly generate files containing crystal information [52, 25]. However, the ability of language models to directly generate files with structural information requires further confirmation, and the generated materials require further verification through DFT calculations.

**Evaluation of Materials Discovery**   The most reliable verification of generated materials is through Density Function Theory (DFT) calculations [53], which uses quantum mechanics to calculate thermodynamic properties such as formation energy and energy above the hull, thereby determining the stability of generated structures [16, 49, 54, 55, 56, 57, 49, 18]. However, DFT calculations require extensive computational resources. Alternative proxy metrics such as pairwise atom distances and charge neutrality [58] were developed as a sanity check of generated materials [15, 25]. Fingerprint distances [59, 60] have also been used to measure precision and recall between the generated set and some held-out test set [61, 62, 13, 25]. To evaluate properties of generated materials, existing works often use a separate graph neural network (GNN) to predict properties of generated material, which is subject to the quality of the property prediction GNN. Furthermore, [63] has shown that although machine learning models can predict formation energies reasonably well, learned formation energies do not reproduce DFT-calculated relative stabilities, bringing the value of learned property based evaluation into question.

## 5   Limitations and Conclusion

We have presented the first diffusion model for materials generation that can scale to train on datasets with millions of materials. To enable effective scaling despite the large number of atoms in complex systems, we developed a novel representation, UniMat, based on the periodic table, which enables any crystal structure to be effectively represented. The UniMat representation is sparse when the chemical system is small, which may incur computational cost that should be reduced by future work. Despite this limitation, we show that UniMat enables training of diffusion models that results in better generation quality than previous state-of-the-art learned materials generators. We further advocate for using DFT calculations to perform rigorous stability analysis of materials generated by generative models.

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

# Appendix

## 6  Architecture and Training

We repurpose the 3D U-Net architecture [64, 65] which originally models the spatial and time dimensions of videos into modeling periods and groups of the periodic table as well as the number of atoms dimension, which can be seen as the time dimension in videos. We apply the spatial downsampling pass followed by the spatial upsampling pass with skip connections to the downsampling pass activations with interleaved 3D convolution and attention layers as in standard 3D U-Net. The hyperparamters in training the UniMat diffusion model are summarized in Table 4.

| Hyperparameter | Value |
|---|---|
| Base channels | 256 |
| Optimizer | Adam ($\beta_1 = 0.9, \beta_2 = 0.99$) |
| Channel multipliers | 1, 2, 4 |
| Learning rate | 0.0001 |
| Blocks per resolution | 3 |
| Batch size | 512 |
| Attention resolutions | 1, 3, 9 |
| EMA | 0.9999 |
| Attention head dimension | 64 |
| Dropout | 0.1 |
| Training hardware | 32 TPU-v4 chips |
| Training steps | 200000 |
| Diffusion noise schedule | cosine |
| Noise schedule log SNR range | [-20, 20] |
| Sampling timesteps | 256 |
| Sampling log-variance interpolation | $\gamma = 0.1$ |
| Weight decay | 0.0 |
| Prediction target | $\epsilon$ |

Table 4: Hyperparameters for training the UniMat diffusion model.

## 7  Details of DFT Calculations

We use the Vienna ab initio simulation package (VASP) [66, 67] with the Perdew-Burke-Ernzerhof (PBE) [68] functional and projector-augmented wave (PAW) [69, 70] potentials in all DFT calculations. Our DFT settings are consistent with Materials Project workflows as encoded in pymatgen [71] and atomate [72]. We use consistent settings with the Materials Project workflow including the Hubbard U parameter applied to a subset of transition metals in DFT+U, 520 eV plane-wave basis cutoff, magnetization settings and the choice of PBE pseudopotentials, except for Li, Na, Mg, Ge, and Ga. For Li, Na, Mg, Ge, and Ga, we use more recent versions of the respective potentials with the same number of valence electrons. For all structures, we use the standard protocol of two stage relaxation of all geometric degrees of freedom, followed by a final static calculation along with the custodian package [71] to handle any VASP related errors that arise and adjust appropriate simulations. For the choice of KPOINTS, we also force gamma centered kpoint generation for hexagonal cells rather than the more traditional Monkhorst-Pack. We assume ferromagnetic spin initialization with finite magnetic moments, as preliminary attempts to incorporate different spin orderings showed computational costs prohibitive to sustain at the scale presented. In AIMD simulations, we turn off spin-polarization and use the NVT ensemble with a 2 fs time step, except for simulations including hydrogen, where we reduce the time step to 0.5 fs.

## 8  Details of AIRSS and Conditional Evaluation

Random structures for conditional evaluation of UniMat are generated through Ab initio random structure search [35]. Random structures are initialized as "sensible" structures (obeying certain symmetry requirements) to a target volume then relaxed via soft-sphere potentials. For this paper, we

always generate 100 AIRSS structures for every composition, many of which failed to converge as detailed in Section 3.3. We try a range of initial volumes spanning 0.4 to 1.2 times a volume estimated by considering relevant atomic radii, finding that the DFT relaxation fails or does not converge for the whole range for each composition. Note that these settings could be further finetuned to optimize AIRSS for convergence rate.

To compute the convergence rate for AIRSS, we use a total of 57,655 compositions from previous AIRSS runs[33], for which 31,917 converged, and hence the AIRSS convergence is 0.55. When we run conditional generation, we randomly sampled 157 compounds from the 31,917 AIRSS-converged compounds, and 309 compounds from the 25,738 compounds where AIRSS had no structure that converged. Among the 157 compounds where AIRSS converged, 137 from UniMat converged, and among the 309 compounds that AIRSS did not converge, 231 from UniMat converged, resulting in an overall convergence rate $137/157 * 31917/(31917+25738) + 231/309 * 25738/(31917+25738) = 0.817$ for UniMat.

## 9 Additional Results

| Method | Dataset | COV-R ↑ | AMSD-R ↓ | AMCD-R ↓ | COV-P ↑ | AMSD-P ↓ | AMCD-P ↓ |
|--------|---------|---------|----------|----------|---------|----------|----------|
| CDVAE | Perov-5 | **99.4** | 0.048 | **0.696** | **98.4** | 0.059 | **1.27** |
| | Carbon-24 | 99.8 | **0.048** | 0.00 | 83.0 | **0.134** | 0.00 |
| | MP-20 | 99.15 | 0.154 | 3.62 | 99.49 | **0.1883** | 4.014 |
| | Perov5 | 99.2 | **0.046** | 0.711 | 98.2 | 0.074 | 1.399 |
| UniMat | Carbon24 | **100** | **0.018** | 0.0 | **96.5** | **0.052** | 0.0 |
| | MP20 | **99.8** | **0.097** | **2.41** | **99.7** | **0.119** | **2.41** |

Table 5: Full proxy coverage metrics from CDVAE. UniMat performs better on larger datasets such as MP-20.

