# OpenReview forum: "Scalable Diffusion for Materials Generation"
_NeurIPS.cc/2023/Workshop/AI4Science — NeurIPS2023-AI4Science Poster_

### Official Review · Reviewer_TfUz · 2023-10-12
**Interesting paper**

**Rating:** 6
**Confidence:** 2

**Review:**

I find the introduction of the innovative UniMat representation for crystal structures quite intriguing. By mapping atom locations to their positions in the periodic table and using diffusion models with interleaved attention and convolution layers, the authors offer a promising alternative to traditional graph-based models. This approach effectively addresses scalability issues in complex chemical systems. I appreciate the rigorous evaluation methods, including both proxy metrics and Density Function Theory (DFT) relaxations, which provide assurance of the physical validity of the generated materials.

I am not an expert in material modeling but I think this direction of modeling large-scale material system can be very promising combined with diffusion modeling.

---

### Official Review · Reviewer_MoRo · 2023-10-20
**LGTM, but needs reviewing by a proper materials scientist**

**Rating:** 7
**Confidence:** 1

**Review:**

This work learns to generate crystal materials using a diffusion process. The diffusion takes place in a novel representation where the coordinates of atoms in the unit cell appear in the corresponding location in a copy of the periodic table. Experiments for conditional and unconditional generation are performed.

In general, the machine learning part of this contribution looks fine to me, but my confidence level on the chemical/materials part is very low: this work should have the approval of at least one referee with a materials science background, and I don't qualify as such.

Some general comments follow.

Do you also apply diffusion to the unit cell parameters $[[a,b,c],[\alpha,\beta,\gamma]]$? I don't see any mention about it in the manuscript, nor what the corresponding prior distribution should be for these parameters.

Lines 102-103 says "$L$ corresponds to the maximum number of atoms per element in the periodic table": did you mean "[...] atoms of the same element per unit cell"?

I am not equipped to properly assess the chemical grounding underpinning this choices leading to the UniMat representation. From a machine learning perspective, UniMat sounds staggeringly sparse, and my intuition would be to seek an invariant representation that has no need for the "data augmentation through random shuffling and rotations" discussed on line 107. I do, however, understand the advantages of this representation for the simple diffusion process it allows to use, plus its compatibility with an adapted U-Net. In the end, I guess the argument is that "it works", but some discussions about the motivations behind this representation could be a nice addition.

---

### Meta-Review · Area_Chair_ttrs · 2023-10-26

**Recommendation:** Accept (Poster)
**Confidence:** 3

**Metareview:**

The authors propose to use the diffusion model to generate material. The presentation is clear and the experimental results are convincing.